# The Socio-Economic Cost of Wind Turbines: A Swedish Case Study

Hans Westlund [1] and Mats Wilhelmsson [2,*]

1  Department of Urban Planning and Environment, Royal Institute of Technology (KTH),
   SE-100 44 Stockholm, Sweden; hans.westlund@abe.kth.se
2  Department of Real Estate and Construction Management, Royal Institute of Technology (KTH),
   SE-100 44 Stockholm, Sweden
*  Correspondence: mats.wilhelmsson@abe.kth.se

**Abstract:** The expansion of wind turbines plays a significant role in developing the ability of a country like Sweden to achieve climate-neutral energy production without relying on nuclear power plants. Wind-turbine energy production is expected to grow in the coming decades. Conflicts may arise between, on the one hand, the government and the energy authority, and, on the other hand, municipalities and property owners, especially if this expansion affects other economic activities, such as tourism and reindeer husbandry, or property values. This report aims to analyse the negative capitalization of wind turbines on property values in Sweden over the last ten years. Our conclusions clearly show a relatively significant capitalization and that this capitalization is relatively local, within eight kilometers of the wind power plant. Large wind turbines, or larger clusters of wind turbines in wind farms, impose a greater socio-economic cost on lower value properties.

**Keywords:** sustainability; wind turbines; capitalization; housing values; hedonic analysis

## 1. Introduction

Global warming has emerged as a threat to the environment and human civilization. The significantly increased $CO_2$ emissions since the 1950s are warming the planet, with consequences that might be disastrous. There is an obvious and urgent need to stop using fossil energy and develop fossil-free energy sources. This need is enhanced by the fact that several countries, among them Japan, Germany, and Sweden, have decided to discontinue their nuclear power plants. Wind power is at present considered the only way to produce the energy needed for this transformation.

It may be expected that local residents would welcome such a transformation to eco-friendly energy production, and consider wind turbines valuable assets that increase an area's attractiveness. However, this is not always the case. Recent news reports detail increasing local resistance to wind power in Germany and Sweden. A German study [1] showed that fear of infrasound from the turbines had the most significant negative influence on the acceptance of wind power, while experiences with wind energy showed a positive effect on acceptance.

It would be easy to characterize the resistance as a Not-In-My-Backyard (NIMBY) phenomenon, i.e., a rejection of necessary societal measures that threaten personal interests, but as pointed out by Wolsink [2–4] and others, e.g., [5–7], this is an overly simplistic explanation. Institutional factors, power structures, social justice, equally distributed regional benefits, and public participation are factors that have been highlighted in the literature.

In Sweden, the resistance seems mainly based on inconveniences such as noise, for those residents within a few kilometers of the wind turbines, as well as altered (deteriorated, destroyed) views during the day and flashing lights from the turbines at night. For the resistance of many permanent residents in peripheral areas, another dimension is added: an

experience of many years of resource exploitation by external actors, first of the forest, then of hydropower, and now also a final resource, the experience of undisturbed nature [8].

The way in which wind power expansion has developed has probably reinforced these feelings. During the initial phase of wind power in Sweden, there were many examples of local stakeholders coming together, sometimes in cooperatives, and building their own wind turbines. The current large-scale expansion of wind power is financed by national and international capital interests, for whom local ownership is uninteresting. As a rule, only a minority of the local population benefits from the direct, positive employment effects and income from land leases [8]. A German study [9] compared two communities in Saxony, in which one had a community-owned wind farm, and the other a commercially owned wind park. Local acceptance of the nearby wind turbines was significantly higher in the community with co-ownership of the wind farm than in the community where a commercial company owned the wind farm.

There are examples of more extensive resistance to wind power among holiday-home owners than among permanent residents (see, for example, Gradén's [10] study of the Swedish province of Dalarna). A British study of planned wind power expansion in the county of Cornwall [11] showed that 95 percent of those who protested against the expansion were residents outside of Cornwall. One explanation to this phenomenon may be that the two groups' local embeddedness differs. Holiday-home owners have their permanent home and livelihood elsewhere, often in an urban environment, and their connection to the area consists in many cases mainly of 'unspoiled' nature experiences and outdoor life. For permanent residents, nature and outdoor life are, of course, also important factors in their local embedding, but permanent residents also have a strong embedding in and dependence on the local economy. Local employment means that more families have jobs, and that the school and the local supermarket survive. These are issues in which the permanent residents are locally embedded but to which holiday-home owners have weaker connections [8].

An argument put forward against wind power expansion is that noise in the immediate area, and deteriorating views for a larger area, would have a negative effect on property values. The international research gives a rather fragmented picture of possible connections between wind power expansion and the development of property values. A number of European studies have found a negative impact of wind farms (or individual turbines) on property values. Jensen et al. [12] found a reduction in house prices of between 5.3% and 15.4% caused by combinations of visual disturbances and noise disturbances in Denmark. In another study, Jensen et al. [13] found that the negative impacts on onshore property values lasted up to 3 km from the wind turbines, while offshore turbines (of which the closest were located 9 km from land) had no impacts. Sunak & Madlener [14] found negatively significant impacts of wind turbines on property prices within the same distance (3 km) in Germany. Sunak & Madlener [15] found that those with a view that from "medium to extreme" was dominated by wind turbines had fallen in price by 9–14%. Still, another German study [16] found an average treatment effect of up to −7.1% for houses within 1 km of a wind turbine, an effect that reached 0 at a distance of 8 to 9 km. A study in the Netherlands [17] found a 1.4% price decrease for houses within 2 km of a turbine. They also found a larger effect for taller turbines. A study in England and Wales [18], based on 1,710,293 property sales within 14 km of wind turbines, found that visible power plants reduced prices by 2.4%.

However, other British studies [11,19] have not found any statistically significant correlations between lower property prices and proximity to wind farms. There are also a number of North American studies that have not found any connection between wind turbines and a change in property prices. Vyn & McCullough [20] studied sales of both single-family homes and agricultural properties in Ontario, Canada, and found no significant impact on either single-family house prices or agricultural property prices. Hoen et al. [21] and Hoen et al. [22] studied the sales of 7500 detached houses within 16 km of 24 different wind farms in the US and found no connections. A third American study by

Hoen et al. [23] studied possible changes in property prices of over 50,000 single-family homes between 1996 and 2011, both before and after construction of wind farms, and found no statistically significant correlations (although there were isolated examples of declining property values). Studies of individual states, such as sales of 48,554 single-family homes in densely populated Rhode Island [24], and 23,000 detached houses and larger, non-plot-divided properties in rural-dominated Oklahoma [25], came to the same conclusions.

There are also several studies showing mixed results. A study of wind farms in northern New York State, USA [26], found a negative correlation between wind farms and property values in two counties but a positive correlation in one county. A study of two Greek islands [27] found a negative impact up to two kilometers from wind turbines on one of the islands, but no effects on the other, probably due to the fact that the second group of turbines was on a very sparsely populated part of that island.

The above conflicting results might suggest a difference between Europe and North America in terms of the possible impact of wind power on property prices, and that the negative effects are mainly found in Europe. It could be that local wind power resistance is stronger in Europe, and that this is reflected in the development of property prices in the vicinity of wind farms or individual power plants. The wind power resistance's argument about falling property prices would in this way become a self-fulfilling prophecy. This hypothesis is supported by Vyn [28], which in a study of Michigan, USA, divides wind power municipalities into those who opposed wind power establishment and those who did not. In municipalities that opposed wind power expansion, he found significant negative effects on property values on properties near wind turbines, while properties within the same radius of wind turbines in municipalities that did not protest did not show any significant price drops. Vyn [28] concluded that an explanation of the relatively large number of studies that found no impact on property values could be that they included areas of both these types.

It can be assumed that studies that found negative impacts on property values would also find that the density of turbines in an area reinforces the negative impacts. However, only a few studies have included turbine density as a control variable, and the assumption is supported by [13,28] but not by other studies [17,26].

This paper takes its starting point in the disparate results described above and analyzes the possible impact of wind turbines/farms on property values in Sweden. While the European countries referred to above are relatively densely populated countries— Denmark 130 inhab./km$^2$; Germany 225 inhab./km$^2$; Netherlands 416 inhab./km$^2$; UK 270 inhab./km$^2$ and Greece 80 inhab./km$^2$—Sweden has only 23 inhab./km$^2$. It could be expected that wind turbines could be located farther away from buildings in sparsely populated Sweden, and thereby have no or at least a weaker impact on property prices than in more densely populated countries. The only study of wind turbines' influence on property values made hitherto in Sweden is a non-peer-reviewed report published in 2010 [29] that did not find any significant negative impact on property prices. However, since the publication of that report, Sweden's wind energy production has expanded nine-fold, and there are thus obvious reasons to investigate the problem again. As the impact of the height and density of wind turbines on property values has been little studied, we analyse these factors in addition to our examination of the impact of distance to wind turbines on property prices.

The remaininder of the article is organised as follows: in the next section, a brief review of wind energy policy in Sweden is reproduced; the third section presents the theoretical platform and the methodological approach that we have used; the fourth section presents the empirical analysis. This is followed by our conclusions and a discussion of policy implications.

## 2. Wind Energy Policy in Sweden

In the wake of the 1973 oil crisis, several experimental wind turbines were set up in the 1970s. However, it was not until the early 2000s that electricity production from

wind turbines increased markedly, especially between 2008 and 2019. In 2019, wind energy produced 20 TWh, which corresponded to 12% of the Swedish electricity production. Sweden's goals to make electricity production 100 percent renewable by 2040, to discontinue nuclear power, and transform production and transportation to be fossil-free require a large increase of wind power electricity.

According to the Swedish Energy Agency [30], at least 100 TWh new electricity must be produced by 2045, which means that current wind power production must increase fivefold. The plan is to allocate new wind farms relatively equally according to the size of geographical regions. Of the 100 TWh, 80 TWh will be produced on land, and 20 TWh offshore. With an estimated average turbine effect of 6 MW, about 1% of Sweden's land area will then be used for wind farms [30].

Energy planning in Sweden takes place at both the national, regional, and local level. According to the Environmental Code, particularly suitable areas for a certain activity in Sweden must be pointed out as a 'national interest'. The designation does not mean that a national interest has been determined, but it signals to courts and public authorities that determine whether such areas are, in fact, a national interest, and give increased weight to the interest. The Swedish Energy Agency is responsible for designating areas of national interest for wind farms. Exceptions are made for national interest areas in national parks, natural coast and mountains, Natura 2000 areas, and nature and cultural reserves.

In 2020, about two-thirds of the Swedish population felt positively about increasing wind power electricity production to provide half of the country's electricity consumption by 2040 [31]. However, at the local level, where wind farms are being planned, there are several examples of strong opposition, both from residents and holiday-home owners. One of the most visible examples of this opposition is renowned footballer Zlatan Ibrahimović, who protested in April 2021 against a wind farm near one of his holiday homes in Jämtland County. https://www.aftonbladet.se/sportbladet/fotboll/a/6zQJ0r/zlatans-nya-fiende-vindkraftverken [In Swedish] (accessed on 10 June 2021).

Except for mineral extraction permits, local municipalities have unilateral authority over development and land use. Municipalities can veto even projects designated for areas of national interest for wind farms. Thus, strong local opposition might influence political decision-makers, preventing the establishment of wind farms.

A large number of articles deal with the socio-economic cost of wind power plant expansion. Most of these articles use the so-called indirect method by analyzing whether wind turbines are capitalised in property values. We have also used this method. Our study's contribution is an analysis of approximately 4000 wind turbines spread throughout Sweden built in the period between 2013–2018. In addition, our results are based on about 100,000 single-family house sales. Another reason why Sweden provides an interesting case study is that Swedish municipalities can veto the continued expansion of wind power. This has made local municipal politicians sensitive to local opinion. As academic studies on this topic have not yet been conducted in Sweden, the results from this analysis will have major policy implications for discussions on institutional arrangements, governance, economic compensation, and wind energy's possible expansion. Methodologically, we have tried to control for endogeneity by estimating the effect within a relatively limited area in the vicinity of wind turbines, and using the propensity score method to estimate a weighted hedonic price model, which, to our knowledge, has not been done before.

## 3. Theoretical and Methodology Framework

Our theoretical starting point is the welfare theory in economics. Climate and climate change can be seen as a classic example of a public good. The cost of another person being able to take advantage of the fact that the climate improvement measure is zero, and climate improvement will not exclude anyone, which is the definition of a public good [32,33]. If society is not met with a sufficient quantity of public goods, it will mean that society as a whole realizes a loss of welfare.

Abandoning fossil-based energy production will lead to reduced greenhouse gas emissions, which in the long run will lead to a better climate for society as a whole and the world. If climate can be considered a (global) public good, where everyone enjoys the benefits, then the measures to achieve climate improvement are a private good with private costs that affect individuals. These costs will not be evenly distributed among all individuals in society, but some individuals may bear a greater private cost than others. Investments in wind turbines are an eco-friendly investment, but at the same time, they give rise to local negative externalities that affect individuals. If these negative externalities are not internalised, the investment (even if it is an eco-friendly investment) will give rise to welfare losses for society. The global benefits to society of a better climate in the future are, of course, great, but the benefits would be even greater if we also compensated those affected by the local negative externalities of the investments [34]. Thus, an important characteristic of eco-friendly investments in wind turbines is that their benefit is global, but their cost is local or national [35].

Our intention here is not to estimate the global benefits of an improved climate, but to estimate the willingness to pay to avoid or accept the local negative externality. Estimating the willingness to pay for an externality can be done in different ways. One method is to directly ask affected individuals about the cost of the negative externality (stated preference) [36]. Another method, often used to estimate the marginal willingness to pay for a negative externality, is to analyse individuals' behaviours in a market (revealed preference) [37]. There are advantages and disadvantages to both methods [38], but we have chosen revealed preference for estimating the willingness to pay, since this case offers a market suitable for analysis, namely the real estate market through the so-called hedonic price method [39]. However, it is important to note that we then only measure the use-value, while stated preference studies measure both use-value and nonuse value [38].

Like many previous studies [11–14,19,20,22–24,26], our analysis is based on the theoretical framework of hedonic models presented by Rosen [40]. There, he showed that under certain conditions the relationship between property prices and the value-affecting attributes could be interpreted as marginal willingness to pay. These value-affecting attributes primarily consist of characteristics that the property possesses, such as size and quality, but different types of amenities are also expected to be capitalised in property values. These can consist of different types of negative and positive externalities and the presence of public goods. One such is, for example, the negative externality of proximity to wind turbines. The hedonic price equation that we will estimate is as follows:

$$HP_{i,t} = \alpha_j + \beta_1 X_{i,t} + \beta_2 WT_{i,t} + \beta_3 T_t + \varepsilon_{i,t} \tag{1}$$

where *HP* is equal to house prices (all models are estimated with price as natural logarithm based on a Box–Cox transformation), and the matrix *X* represents all value-affecting attributes such as size, age, and location. The variable *WT* represents proximity to a wind turbine. In the empirical analysis, we either used proximity to a wind turbine as a binary variable, or the shortest distance to a wind turbine. The vector *T* is a binary variable measuring the month the property was sold (fixed time effects). The subscripts *i* and *t* indicate transaction and time. All Greek letters indicate parameters that are estimated. The parameter $\alpha$ has a subscript of *j* for a municipality, indicating that fixed regional effects are included in the model.

The parameter estimate for wind turbines is the implicit or hedonic price and is interpreted as the marginal willingness to pay. In a cost–benefit analysis, this socio-economic cost must be set against the socio-economic benefits that a fossil-free energy source creates. The purpose here, however, is to estimate the capitalization effect of wind turbines on property values.

One of the most serious problems in this type of analysis is the issue of endogeneity. Here, there is a risk that wind turbines have been located in areas that are less attractive and thus have lower property values. The parameter estimates of being located near a wind farm will then obviously show a negative correlation. An argument against this

reasoning could be that wind turbines have not primarily been located in areas with lower property values, but areas with primarily good wind conditions. However, there is also a risk that they are located where they have the least impact on housing.

A few different methods reduce the risk of detecting a spurious relationship. Perhaps the most common is to use different instrument variables [41], and another is the so-called difference-in-difference methodology [13,33]. Here, we have not used any of these, but instead the propensity score method [42] to identify properties similar in size and location but not in proximity to wind turbines. Thereby, the intention is to reduce the endogeneity problem, even if we cannot completely ignore it. We have also estimated models within a relatively narrow distance from the wind turbines to minimise the risk that properties near the wind turbines are different from properties further away. It will also reduce the risk of spatial dependence. We have also included fixed municipal effects, distance to urban areas, longitude, and latitude in the hedonic price equation to reduce the risk of spatial dependence [43].

In the first step, we have estimated a logistic regression model, where the dependent variable consists of a binary variable that indicates whether the property is close to the wind turbine. For each property, we have then estimated a probability that they are located near a wind turbine. These probabilities have then been used as weights (the inverse of the probability) in the second step when we have estimated the hedonic price equation. Thus, properties with a high probability of being close to a wind turbine will be weighed heavier in the regression analysis than those that are further away, regardless of whether they are close to the wind turbine or not. The propensity score method has been used in this way in, for example, [44].

$$HP_{i,t} = \alpha_j + \beta_1 \frac{X}{PS}_{i,t} + \beta_2 \frac{WT}{PS}_{i,t} + \beta_3 T \frac{1}{PS_t} + \varepsilon_{i,t} \tag{2}$$

where *PS* is the estimated propensity score. The higher the probability that the property is similar to the properties close to wind turbines, the greater weight the observation will have in the estimate.

*Parameter Heterogeneity*

Proximity to wind turbines has been calculated by the shortest Euclidian distance between a wind turbine and the property. However, very rarely are wind turbines located in isolation. Instead, they are located in wind farms with two or more wind turbines relatively close together. To obtain a better estimate of the capitalization effect, we have identified wind power plants close to each other through cluster analysis. In addition, we have analyzed whether the size of the wind turbines has an impact on the capitalization effect by seperating those properties close to wind turbines that are larger than average from those that are close to wind turbines taller than average.

## 4. Empirical Analysis

The empirical analysis is divided into four different subsections. First, the data used are presented, then the sample weights are estimated using the propensity score methodology. This is followed by the estimation of several hedonic price equation models. The section concludes with an analysis of the effect of wind turbine size on the capitalization effect.

Results from what we term the 'full' and 'reduced' samples are presented throughout. The full sample includes all real estate transactions that have taken place throughout Sweden, while the reduced sample includes only real estate transactions that occur within 20 km of the wind power plant. We have reduced the sample because our basic hypothesis is that the capitalization effect is relatively local, partly as a test of the robustness of our results.

### 4.1. Data and Variables Used

We use two data sources. First, we utilise sales data regarding single-family houses provided by the company Mäklarstatistik AB; second, we use data regarding the location of wind turbines in operation from the Swedish Energy Agency.

The data is based on sales transactions provided by several member companies of one of Sweden's largest property agents. The coverage rate of sales is good overall, at around 80–90 percent, but slightly better in urban than rural areas. Wind turbines are primarily rural, so there is a risk that the number of analyzed sales is slightly lower than the total number of sales. There is a risk of selection bias, but it should be limited to the cheapest properties not included in the analyzed data. The available information includes when the property was sold, the transaction price, living space, and plot area, detached or semi-detached house, and latitude and longitude coordinates.

The information from the Swedish Energy Agency consists of each wind turbine in operation in Sweden, its location (latitude and longitude), its height, and its energy capacity. The calculation of the shortest distance between property and wind turbine has been done similarly to that of Heintzelman and Tuttle [26]. Hence, we calculate the shortest Euclidean distance to a wind turbine for each property.

In addition to this information, we have also used information about 1600 urban areas in Sweden and calculated the shortest distance between a property and an urban area. An urban area, in this study, is defined as a collection of buildings where the distance between the properties is less than 200 meters, and with a population greater than 200 (in the cohesive community). Furthermore, the daytime population must be at least 10 percent greater than the resident population. Descriptive statistics for the two data sets are given in Table 1.

**Table 1.** Descriptive statistics.

| | Full Sample | | Restricted Sample | |
|---|---|---|---|---|
| | **Mean** | **Standard Deviation** | **Mean** | **Standard Deviation** |
| Price | 3,079,324 | 2,135,857 | 2,717,670 | 1,657,671 |
| Living area | 129.5 | 36.0 | 129.0 | 35.8 |
| Age | 52.2 | 28.5 | 52.7 | 29.2 |
| Rooms | 5.2 | 1.3 | 5.1 | 1.3 |
| Rowhouse | 0.08 | 0.3 | 0.07 | 0.2 |
| Semidetached | 0.06 | 0.2 | 0.05 | 0.2 |
| Distance to urbanization | 2.9 | 2.4 | 2.6 | 2.2 |
| Distance wind turbine | 14.9 | 11.4 | 8.8 | 5.4 |
| No. of obsevations | 97,229 | | 69,941 | |

The number of single-family house sales has been divided into a full sample (97,229 transactions) and a reduced sample (68,941 transactions), where the reduced sample includes only sales within 20 km of a wind power plant. The descriptive statistics show that the dependent variable transaction price amounts to just over SEK 3 million in the full sample, compared with approximately SEK 2.7 million in the reduced sample. The lower purchase price indicates that properties closer to wind turbines are affected by the proximity. However, the causal relationship here can be questioned. A bit surprising is that properties within 20 km of wind turbines are also nearer urbanised areas, which is expected to increase property values. The size of the properties are almost identical, in terms of living space, the number of rooms, and the plot area. The two samples are also equivalent in terms of property age. One difference of note is that terraced houses are more common in urbanised areas, and thus represent a greater proportion of the full sample than the reduced sample.

The average property distance to the nearest wind turbine is almost 15 km in the full sample, and almost nine kilometers in the reduced sample. In total, just over 4 percent of properties are within two kilometers of a wind turbine. As expected, the share is slightly

higher in the reduced sample, but only 6 percent of the total number of transactions are included in the reduced sample. We have used two kilometers, as it has been used previously in the literature, including by Skenteris, Mirasgedis, and Tourkolias [27], who use a range of 0–2 km, and [23], which uses 1 mile (approximately 1.6 km). Unlike Heintzelman and Tuttle [26] and others, we have relatively many properties close to wind turbines (almost 4000 properties are located within two kilometers of a wind turbine).

In total, we analyze the impact of 4337 wind turbines on property values. These wind turbines are spread throughout Sweden, but are most numerous in the municipalities of Piteå, Strömsund, Gotland, and Örnsköldsvik. All analyzed wind turbines are land-based. Furthermore, they are located at an average of 342 m above sea level (with a standard deviation of 210 m), and the average height of the wind turbines is 173 m, with a standard deviation of 44 meters. Approximately 25 percent of wind turbines have a height above the average height, which means that many wind turbines are significantly smaller than the average. The completed cluster analysis shows that only 100 wind farms are found to have more than ten wind turbines.

*4.2. Propensity Score Estimates*

In the first step, we have calculated each property's probability to be close to a wind power plant using the so-called propensity score method. As said earlier, the intention is to reduce the endogeneity problem by only analyzing properties that are similar in size and location but not in proximity to wind turbines. In principle, this means that we have estimated a logistic regression where the dependent variable is whether the property is within two kilometers of the wind power plant. As dependent variables, we have used property attributes such as dwelling size in square meters, and the number of rooms, size of plot, property age, and semi-detached or row house. In addition, we included fixed effects for counties and for when the property was sold. Exact longitude and latitude coordinates are also included in the model. It has been shown that the inclusion of coordinates can remedy the problem of potential spatial dependency [13,16,42].

Table 2 shows the results of the logistical regression. In the first model, we have used all properties during the period, and in model two, only the properties within 20 km of the wind turbine are included. Figures 1 and 2 show the distribution of propensity score weights in the full sample and the restricted sample.

The dependent variable consists of whether the property is up to two kilometers from the wind turbine, i.e., about 4–6 percent of the sample. Larger dwellings (measured by living area and number of rooms) seem to lower the probability that the property is close to wind turbines, while plot size increases the probability. This means that properties closest to wind turbines are smaller indoors, but include larger tracts of land. We can also note that property age has a positive effect on the probability, which indicates that properties closest to wind turbines are older, and probably pre-exist the wind turbines. Townhouses and rowhouses are usually not located within two kilometers of wind turbines. Increased distance to urban areas (also highly correlated with the presence of terraced and row houses) reduces the probability of proximity to a wind turbine, i.e., wind turbines have not been located close to pre-existing urban areas. The differences between the two selections are small. The models also include fixed effects for the county and date of property sale. The degree of explanation is relatively low, which is normal for a logistic regression. The logistic regression results in propensity scores that will then be used as regression weights in the hedonic price model. Figure 1 shows the distribution of these weights in the full sample and the reduced sample.

The weights range from 0 to 1. The closer to 0 a sale falls, the less weight the observation will have in the model that estimates the capitalization effect. Hence, the lower the weight, the worse the observation as a comparative transaction. However, it is important to remember that all observations will be included in the hedonic price model, but some will have a higher or lower weight in the regression (weighted hedonic regression model).

**Table 2.** Propensity score model (logistic regression).

|  | (1) | (2) |
|---|---|---|
|  | **Full Sample** | **Restricted Sample** |
| Living area | −0.00112 | −0.00108 |
|  | (−1.83) | (−1.75) |
| Lot size | 0.0000523 *** | 0.0000520 *** |
|  | (8.17) | (8.06) |
| Age | 0.00485 *** | 0.00489 *** |
|  | (9.55) | (9.72) |
| Rooms | −0.0447 ** | −0.0429 * |
|  | (−2.65) | (−2.54) |
| Rowhouse | −0.382 *** | −0.375 *** |
|  | (−4.88) | (−4.78) |
| Semidetached | −0.342 *** | −0.367 *** |
|  | (−4.05) | (−4.34) |
| Distance to urbanization | −0.0430 *** | −0.0420 *** |
|  | (−5.29) | (−5.19) |
| Longitude | −0.311 *** | −0.297 *** |
|  | (−5.87) | (−5.48) |
| Latitude | 0.0384 | 0.0346 |
|  | (1.26) | (1.15) |
| Constant | 12.35 *** | 13.27 *** |
|  | (3.93) | (4.14) |
| N | 97,161 | 68,909 |
| $R^2$ | 0.1067 | 0.0579 |
| AIC | 31,522.8 | 30,389.2 |

The model also includes fixed municipal effects, fixed monthly effects and latitude and longitude. $t$ statistics in parentheses. * $p < 0.05$, ** $p < 0.01$, *** $p < 0.001$.

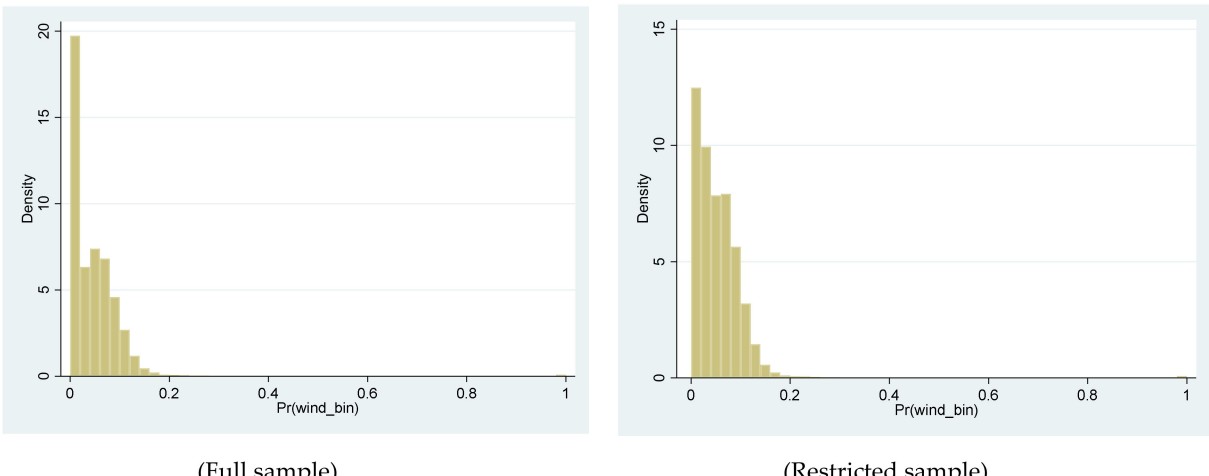

(Full sample)  (Restricted sample)

**Figure 1.** Histogram Propensity Scores.

### 4.3. Hedonic Price Equation

In the second step, we have estimated the hedonic price equation using weighted least square regression (WLS), where the inverse of the propensity score is the weights. Table 3 presents four models. Models 1 and 2 include the distance to wind turbines measured in kilometers, i.e., the variable is continuous. Model 1 uses all transactions, and model 2 is based on the restricted sample. In models 3 (full sample) and 4 (restricted sample), the variable proximity to wind turbines is instead measured as a binary variable, equal to 1 if the property is within two kilometers of the wind turbine.

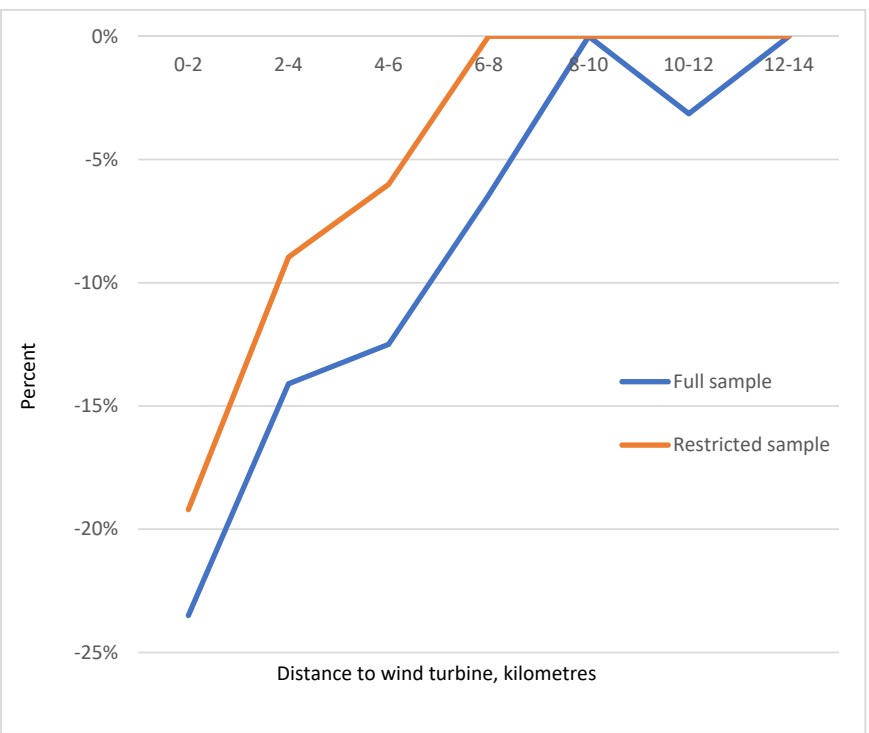

**Figure 2.** Capitalization effect.

In general, we have a relatively good degree of explanation in our models. The price variation can be explained better in samples with all sales compared to the reduced sample. This indicates more heterogeneity among the sales found in the reduced sample. In the full sample, the degree of explanation amounts to as much as 82 percent, which can be considered very high; in the reduced sample, the degree of explanation amounts to almost 70 percent, i.e., a relatively large difference in degree of explanation. Compared with Heintzelman and Tuttle [26] and Skenteris, Mirasgedis, and Tourkolias [27], the degree of explanation is very high, and compared to Hoen et al. [23], the degree of explanation in this study is in the upper range.

All estimated parameters show expected signs and a reasonable level compared to other studies. The size of the house, as measured in square meters, number of rooms, and plot size, increases expected prices, while if the property is a terraced house or chain house, expected values decrease. For example, if the number of rooms increases by one room, the price is expected to increase by about 3 percent, and if the property is a row house, the value is reduced by 22 percent in the full sample, and halved in the reduced sample. Longer distances from urban areas reduce the value of the property, as expected.

Proximity to wind turbines has a clear and statistically significant effect on property values. In economic terms, the effect is relatively significant. Greater distances from wind turbines increase the expected price of the property. In the full sample, the effect is twice as great as in the reduced sample, which is probably due to a certain endogeneity resulting from wind turbines' location in areas with lower housing prices. When we analyze a narrower area around the wind turbines than that found in the reduced sample, the effect is lower and more reliable. The economic interpretation of the estimated parameter is that every additional kilometer between the property and the wind power plant increases the housing value by 0.3 percent. Here, we have estimated the impact as a linear impact, which it certainly is not. We are expected to have a higher impact nearer the wind turbine than further away.

**Table 3.** Hedonic price equation with PS weights (continuous and binary variables).

| | (1) | (2) | (3) | (4) |
|---|---|---|---|---|
| | **Full Sample** | **Restricted Sample** | **Full Sample** | **Restricted Sample** |
| Living area | 0.00396 *** | 0.00471 *** | 0.00397 *** | 0.00471 *** |
| | (58.01) | (48.65) | (57.45) | (48.80) |
| Lot size | 0.0000111 *** | 0.0000165 *** | 0.00000846 ** | 0.0000169 *** |
| | (4.70) | (10.02) | (2.94) | (10.36) |
| Age | −0.00179 *** | −0.00307 *** | −0.00169 *** | −0.00305 *** |
| | (−22.16) | (−28.78) | (−20.89) | (−28.70) |
| Rooms | 0.0368 *** | 0.0268 *** | 0.0374 *** | 0.0266 *** |
| | (19.30) | (10.49) | (18.83) | (10.43) |
| Rowhouse | −0.220 *** | −0.111 *** | −0.221 *** | −0.110 *** |
| | (−49.16) | (−15.38) | (−49.45) | (−15.46) |
| Semidetached | −0.149 *** | −0.0922 *** | −0.153 *** | −0.0918 *** |
| | (−30.36) | (−11.75) | (−29.98) | (−11.75) |
| Distance to urbanization | −0.00965 ** | −0.0168 *** | −0.00483 | −0.0166 *** |
| | (−2.88) | (−9.32) | (−1.02) | (−9.48) |
| Distance to wind turbine | 0.00775 *** | 0.00333 *** | | |
| | (7.55) | (4.23) | | |
| Wind turbine (binary) | | | −0.149 *** | −0.141 *** |
| | | | (−9.21) | (−14.56) |
| Constant | 14.31 *** | 14.48 *** | 14.45 *** | 14.55 *** |
| | (209.50) | (151.36) | (226.14) | (153.54) |
| N | 97,161 | 68,909 | 97,161 | 68,909 |
| $R^2$ | 0.820 | 0.690 | 0.818 | 0.691 |
| AIC | 44,841.4 | 49,144.0 | 46,135.2 | 48,931.9 |

The model also includes fixed municipal effects, fixed monthly effects, latitude, and longitude. $t$ statistics in parentheses. ** $p < 0.01$, *** $p < 0.001$.

If we estimate the capitalization effect using a binary variable (where 1 refers to whether the property is within two kilometers of a wind turbine, otherwise 0), it can be stated that the effect is significantly higher. We can also note that the effect is equivalent regardless of whether we analyze all transactions or only the reduced sample. The effect here is approximately of the order of 14 percent lower value.

If we compare our results, we find that they are in line with many other studies from other countries, such as Skenteris, Mirasgedis, and Tourkolias [27], although our results diverge from the estimates of others, such as Hoen et al. [23]. Studies that have found significant effects have drawn criticism for basing their results on small samples and relatively few wind turbines, where only a few properties are located near the wind turbines. This is, among other things, an argument put forward by Hoen et al. [23]. However, our sample (regardless of whether we analyze the full sample or the reduced sample) is large, and has a significant number of properties located in the vicinity of a wind turbine.

Table 4 illustrates WLS estimates of the hedonic price equation where the proximity to wind turbines consists of several binary variables, where the first refers to the range 0–2 km, the second 2–4 km, etc. That is, we are relaxing the assumption about linear capitalization. It is also a specification used by, for example, Heintzelman and Tuttle [26]. As before, model 1 refers to all transactions, and model 2 refers to the restricted sample. Figure 2 shows the capitalization effect within the interval of 0–14 km based on the estimates in Table 4.

**Table 4.** Hedonic price equation with PS weights (structure of binary variables).

|  | (1) | (2) |
|---|---|---|
|  | **Full Sample** | **Restricted Sample** |
| Living area | 0.00396 *** | 0.00470 *** |
|  | (57.63) | (48.74) |
| Lot size | 0.00000883 ** | 0.0000173 *** |
|  | (3.17) | (10.55) |
| Age | −0.00167 *** | −0.00302 *** |
|  | (−20.72) | (−28.43) |
| Rooms | 0.0373 *** | 0.0261 *** |
|  | (18.93) | (10.25) |
| Rowhouse | −0.222 *** | −0.112 *** |
|  | (−48.98) | (−15.50) |
| Semidetached | −0.153 *** | −0.0901 *** |
|  | (−30.00) | (−11.52) |
| Distance to urbanization | −0.00553 | −0.0169 *** |
|  | (−1.24) | (−9.21) |
| Wind turbine 0–2 km | −0.235 *** | −0.192 *** |
|  | (−10.41) | (−12.61) |
| Wind turbine 2–4 km | −0.141 *** | −0.0896 *** |
|  | (−6.14) | (−6.01) |
| Wind turbine 4–6 km | −0.125 *** | −0.0601 *** |
|  | (−5.67) | (−4.24) |
| Wind turbine 6–8 km | −0.0645 ** | −0.0220 |
|  | (−3.28) | (−1.63) |
| Wind turbine 8–10 km | −0.0127 | 0.00121 |
|  | (−0.83) | (0.10) |
| Wind turbine 10–12 km | −0.0314 * | 0.00485 |
|  | (−2.29) | (0.40) |
| Wind turbine 12–14 km | −0.0175 | 0.0160 |
|  | (−1.26) | (1.32) |
| Wind turbine 14–16 km | −0.0780 *** | −0.0524 *** |
|  | (−5.94) | (−4.00) |
| Wind turbine 16–18 km | −0.0541 *** | −0.0445 *** |
|  | (−5.00) | (−3.77) |
| Constant | 14.45 *** | 14.54 *** |
|  | (225.04) | (152.56) |
| N | 97,161 | 68,909 |
| $R^2$ | 0.819 | 0.693 |
| AIC | 45,612.8 | 48,525.6 |

The model also includes fixed municipal effects, fixed monthly effects, latitude, and longitude. *t* statistics in parentheses. * $p < 0.05$, ** $p < 0.01$, *** $p < 0.001$.

In the models where proximity to wind turbines is included as several binary variables, the degree of explanation is on a par with previous models. Additionally, all underlying variables have the same sign and magnitude. This means that regardless of how wind turbines are included in the model parts, other estimates are robust. However, an exception is proximity to urbanization, which is not statistically significant in the model where all observations are included, but in the model with the restricted sample, the estimate is on a par with previous estimates.

The effect of wind turbines is also clear in this model, and it is also clear that the effect is non-linear. For every kilometer from the wind turbine, the marginal effect is lower. Within the range 0–2 km, the effect is greatest. Here, the estimated capitalization effect is approximately 19–23 percent. This is a significant effect, and greater than the estimates (ref) of many other studies. The estimate is statistically significant, but it should be noted that the estimate is based on relatively few observations. Few properties are in the range of 0–2 km. In the interval 2–4 km, the estimated effect is 10–14 percent, and then drops to 6–12 percent in the interval 4–6 km. In the interval 6–8 km, the effect falls to 2–6 percent.

Distances greater than 8 km do not appear to have statistically significant estimates, nor does the effect appear to recur at distances greater than 14 km. This effect is difficult to explain. The greater the distance to the wind turbine, the closer it is to other 'disamenities' that give a negative capitalization but are not included in the model. However, the results show that further research is needed to understand the wind power plants' capitalization in property values. The result of the capitalization in the interval of 0–10 km is shown in Figure 2.

### 4.4. Height and Number of Wind Turbine

In step 4, we have analyzed whether the size of the wind turbines has any significance for the capitalization effect. We have estimated two models to estimate the capitalization effect when the property is close to large wind turbines or small wind turbines. Moreover, we have also estimated the capitalization effect when the property is located near larger wind farms. The expected effect is that wind turbines taller than average have a greater capitalization effect, and that wind farms with more than 10 wind turbines have a greater capitalization effect. Table 5 illustrates the WLS estimates regarding the effect of height and number of wind turbines.

If we divide the material into properties proximate to higher than average wind turbines, we can first state that relatively few properties are within 20 km of these. The heterogeneity also seems to be greater, as the degree of explanation drops to just over 60 percent. The estimated effect of being in the range of 0–2 km, compared to 18–20 km, is statistically significant, and the economic interpretation is that the effect is significant—just over 40 percent lower prices, according to the model.

Compared with the model with only the properties proximate to shorter wind turbines, we have significantly more sales in the interval 0–20 km from the wind turbine. The degree of explanation is also higher in this model. The effect of being a maximum of 2 km from the wind power plant is significantly lower, at just under 10 percent. The interpretation is that the authorities need to be much more careful where they locate the newer, much taller wind turbines than they may have been previously. The effect on property owners will be significantly greater.

In the model where we analyse housing prices for properties close to wind farms, the effect is almost 30 percent. Although the estimate is based on a smaller number of observations, the estimates are statistically significant. The marginal effect of proximity to tall wind turbines, or wind farms, should be studied more carefully to minimise impact on property values when situating future wind turbines or establishing compensation levels for affected property owners.

**Table 5.** Parameter heterogeneity in size (altitude and number). Restricted sample.

| | (1) | (2) | (3) |
|---|---|---|---|
| | **Large** | **Small** | **More** |
| Living area | 0.00500 *** | 0.00462 *** | 0.00533 *** |
| | (8.76) | (51.93) | (10.56) |
| Lot size | 0.00000965 | 0.0000174 *** | 0.0000412 *** |
| | (1.93) | (10.36) | (6.11) |
| Age | −0.00365 *** | −0.00218 *** | −0.00473 *** |
| | (−5.27) | (−21.13) | (−6.03) |
| Rooms | 0.0205 | 0.0290 *** | 0.0242 |
| | (1.27) | (11.95) | (1.53) |
| Rowhouse | 0 | −0.131 *** | −0.110 |
| | (.) | (−16.65) | (−0.87) |
| Semidetached | −0.176 | −0.0937 *** | −0.251 *** |
| | (−1.76) | (−13.58) | (−3.62) |

**Table 5.** *Cont.*

|  | (1) | (2) | (3) |
| --- | --- | --- | --- |
|  | **Large** | **Small** | **More** |
| Distance to urbanization | −0.00944 | −0.0164 *** | −0.0351 *** |
|  | (−1.12) | (−10.44) | (−4.37) |
| Large wind turbine (binary) | −0.412 *** |  |  |
|  | (−7.45) |  |  |
| Small wind turbine (binary) |  | −0.0985 *** |  |
|  |  | (−13.51) |  |
| Wind turbine park (binary) |  |  | −0.277 *** |
|  |  |  | (−3.51) |
| Constant | 30.42 | 63.07 *** | 27.46 |
|  | (1.18) | (18.64) | (1.23) |
| N | 1054 | 68,178 | 1845 |
| R$^2$ | 0.620 | 0.723 | 0.442 |
| AIC | 779.8 | 41,912.3 | 1476.4 |

The model also includes fixed municipal effects, fixed monthly effects, latitude, and longitude. *t* statistics in parentheses. *** $p < 0.001$.

## 5. Conclusions and Policy Implications

Wind turbines are one of many eco-friendly investments necessary to ensure a climate that makes the planet habitable. The benefits of these climate-improvement measures can be seen as global public goods that everyone on earth will benefit from. However, the investment itself is a private good with local negative externalities. These negative externalities also give rise to a loss of welfare for society if they are not internalized in any way. Furthermore, the cost is unevenly distributed among regions and individuals. In order for it to be Pareto-optimal, those who bear the cost must be compensated. Our study should be seen as providing guidance for locating wind turbines with minimal local socio-economic costs, or for enabling rational compensation of effected individuals and households. Since we have only valued use-value, there may be an underestimation of the total local social cost of the investment in wind turbines.

The results clearly indicate a negative capitalization of proximity to wind turbines on property values in Sweden. The relationship between wind turbines and property values is non-linear and decreases exponentially with the distance from the wind turbines. The results also indicate that proximity to tall wind turbines and proximity to many wind turbines (wind farms) have greater impacts.

As Sweden plans to increase its wind power production fivefold in the next two decades, these results will doubtless have policy implications. Even if protests against wind power expansion remain at the local level, the expansion is likely to lead to more and better organised protests. It can also be expected that property owners will demand economic compensation for decreased property values. All this indicates the need for a national policy, not only for expanding wind power production (which is underway), and possibly abolishing municipalities' opportunity to veto against planned wind power establishments (currently being investigated), but also for handling individual demands for compensation and local fears of the eyesore presented by nearby wind parks. Currently, a governmental investigation is analyzing the possibility of abolishing municipalities' opportunity to veto against planned wind power establishments. As shown by international research, such a measure would probably strengthen local stakeholders' feelings of powerlessness and reduce their trust in society's institutions. This is a theme outside the scope of this article, but a very important topic for future research.

Regarding property values, future research could address the endogeneity problem with a difference-in-difference approach. Wind turbines have been built at different times, so an analysis of before and after construction can be calculated, even if it might be difficult to find fully alternative reference locations. Information about the entire construction process, from building permits, construction, and operation, could also be used to analyze the project's capitalization effects. Data about rejected building permits are also interesting to further analyze. Another possible topic for further research are the regional or other locational differences in capitalization.

The significance of this type of study will become increasingly important. The policy implications are clear. Wind turbine energy production has expanded in recent years, and will certainly continue to expand to meet the goal of climate-neutral energy production. To gain acceptance for continued expansion, values beyond environmental values, including property values, must be considered when wind turbines are built. Further research can form the basis for calculating compensation to property owners.

**Author Contributions:** Conceptualization, H.W. and M.W.; methodology, H.W.; M.W.; software, M.W.; validation, H.W.; M.W.; formal analysis, M.W.; investigation, H.W.; M.W.; resources, M.W.; data curation, M.W.; writing—original draft preparation, H.W.; writing—review and editing, H.W.; M.W.; visualization, M.W. Both authors have read and agreed to the published version of the manuscript.

**Funding:** This research received no external funding.

**Institutional Review Board Statement:** Not applicable.

**Informed Consent Statement:** Not applicable.

**Conflicts of Interest:** The authors declare no conflict of interest.

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
