# Peer review of "The Socio-Economic Cost of Wind Turbines: A Swedish Case Study"

_sustainability, doi:10.3390/su13126892_

Round 1

Reviewer 1 Report

I believe that this paper deals with quite an interesting topic of how an environment-friendly technology may face severe resistance from people who would eventually enjoy the goodness of environment-friendly technology.  Most of all, the authors have made great efforts to collect and arrange the data, constructed a relevant estimation model in which house price is regressed by the geographical proximity to the wind turbines, and conducted a very refined statistical analysis. That is, it seems obvious that the authors are very knowledgeable about the econometric methodology. However, while they have somehow shown their method capabilities, it seems to me that the storytelling is much lower than expectations, which neither properly follows the conventions of the academic empirical study, nor effectively emphasizes the potential contributions of their research. Thus, I would like to give my critical suggestions that require major revision to a significant degree as follows.

It seems that the assertion that the study is the first study examining the negative impacts of wind turbines on the public’s concerns of their properties in Swedish context is the only contribution that the authors claim. As a paper pursuing the publication of a journal having global audiences, a study of a local context is unlikely to draw the attentions from the potential audience. For sure, there have always been the local people’s severe resistances to the government-level attempts to build eco-friendly facilities globally everywhere, and the decrease of the real estate property values is a main concern of the people who resist, so such a situation would trigger a debate of whether the eco-friendly facilities would actually or consequently increase or decrease the value of the property near to the facilities. Why didn’t the author highlight such debate and their actual examination about it? I would like to recommend the authors to do the major revision especially on the introduction section in such a direction. It’s unlikely at least for me to try to know only about Sweden, but I would read a paper that tries to argue about a bigger, broader, and more general concerns. 

If the introduction is revised in a manner to highlight a general research question, the literature review section, i.e., 3. Theoretical and methodology framework, must be revised so. That is, the authors have to focus on the literature that deals with the public’s resistances to the government’s eco-friendly policies and finds the concerns of the value fall of their properties. Even if the empirical test doesn’t directly address the motivation of the public’s value concerns, the authors had better show their theoretical efforts to address why people are concerned of the decrease of their direct economic benefits, rather than the increase of the common’s social and environmental benefits. I would like to find such efforts of the authors in the revised manuscript. Therefore, the section 3’d better be renamed “literature review”, other than “theoretical and methodology framework.” All the estimation models introduced as mathematical formulae are therefore replaced in the following “Empirical Analysis” section. 

A separate section in which the measures of all the variables included in the estimation models are specially introduced is required. In the current manuscript, the description of the estimation models, the variables, and the findings are all very messy, so the readers cannot easily follow what the authors have exactly done. I would have followed the conventions of the empirical papers in which the method section proceeds from data, measurement, estimation, etc. The readers would think “Wow, the authors did something, but I don’t know what it is” in the current manuscript. I would suggest that the authors consider the reader friendliness first.  As such, the explanations about why the estimations must proceed in two different steps must be given to the authors. And why the authors provide full sample model as well as restricted sample model must be more easily found and understood by the readers. I know all the analyses had to be done to examine the proposed effects of the proximity to turbines on the house price, but it would be much better if each analytical step is well described and highlighted in terms of its usages and necessities. 

The amount of the conclusion and policy implication section in page 14 must be enriched as much as the introduction sections. Please be friendly to the readers by double-emphasizing the contributions of the papers, giving the interpretations of the findings and the potential applications of it, and confessing about the potential limitation of the paper both theoretically and methodologically. The value of the paper must be found from the thoughts developed by the authors from all the processes of studies, not from the technical skills of empirical analysis. I’d suggest the authors to assume the readers who are not knowledgeable about the empirical methods of the paper, but very interested in the story of why people sometimes do not like the eco-friendly policies and facilities.

Author Response

Reply:

We thank the reviewer for these important comments. We can only agree that the storytelling of the former version was poor. Now, we have completely rewritten the two first sections and added much more information and references.

Regarding the motivation for the study of Sweden, we have now added the fact that Sweden is a sparsely populated country compared with most other European countries. This could mean that wind turbines would not have any (or at least lower) impacts on property values. On the contrary, our results show negative impacts at a longer distance than most previous research. Another reason why Sweden provides an interesting case study is that Swedish municipalities can veto the continued expansion of wind power.

Regarding the issue why not eco-friendly facilities increase property values, we have added a discussion on reasons to the local resistance, with several examples from the literature. We have added a discussion on why local resistance to wind turbines exists and we show that the reasons go far beyond the NIMBY argument. That is, protesters are not against eco-friendly energy, but they are against disturbing sounds, changed (destroyed) views, how wind power expansion is handled, and not least local peoples’ lack of power and influence. We conclude that this very well can result in a self-fulfilling prophecy and make the areas in question less attractive on the property market.

In Section 3, we have also added a theoretical explanation:

“If climate can be considered a (global) public goods where everyone enjoys the benefits, then the measures to achieve climate improvement are a private good with private costs that affect individuals. These costs will not be evenly distributed among all individuals in society, but some individuals may bear a greater private cost than others. Investments in wind turbines are an eco-friendly investment, but at the same time, they give rise to local negative externalities that affect individuals. If these negative externalities are not internalised, the investment (even if it is an eco-friendly investment) will give rise to welfare losses for society. The global benefits to society of a better climate in the future are, of course, great, but the benefits would be even greater if we also compensated those affected by the local negative externalities of the investments [34]. Thus, an important characteristic that eco-friendly investments in wind turbines have is that the benefit is global, but the cost is local or national [35].”

In section 4, we have included a new introduction to the empirical analysis. Hopefully it is more clear now.

Reviewer 2 Report

The research topic is very good and right on time. The abstract is well organized, However, the introduction and literature review are quite short and could be improved. The theoretical and methodolody framework quite well. Results, tables, and figures are well discussed and labeled. However, the discussions should be well connected to the results. The conclusion is precise and the references are most recent but very few, could be imporoved. Also, consider having minor corrections on English language and styles.

Author Response

Reply: We have completely rewritten the two first sections, developed the presentation and added a large number of new references. The English has been checked by an authorized translator.

Reviewer 3 Report

The conclusions of the presented study are in order and they could be expected, as the authors state. Statistical confirmation and explanation are sufficient.

In the discussion, it is necessary to compare the above results with studies aimed at declining real estate prices also due to other externalities, e.g. other types of power plants or industrial enterprises, presented by the authors in several journals registered in WOS and Scopus. The addition of comparison with other countries would enrich the contribution presented. It is appropriate to supplement the literature search with other relevant sources, in order to strengthen the importance of research and its conclusions. If the authors provided possible solutions to ensure the reduction of the negative effects of wind turbines on the population and possible health problems, which are mentioned from an interdisciplinary point of view by other scientists researching wind power plants, it would be beneficial for the importance of the contribution and its impact.
The used method of citations in the text should be modified by default, e.g. for point 11, it is necessary to list the authors for the chosen method of formulation.

Author Response

Reply: Regarding possible solutions, the conclusion we can draw from this study is that economic compensation for reduced property values is one important component in reducing the negative effects. We also refer to other studies, saying that institutional factors, power structures, social justice, equally distributed regional benefits, and public participation are factors that must be considered in forming a policy for reducing the negative social effects for affected individuals.

Round 2

Reviewer 1 Report

All my concerns are well addressed in the revised manuscript, so I have no further comments. Well done. 

Author Response

Thank you! Great comments!